# Extracellular Vesicles in Liquid Biopsies as Biomarkers for Solid Tumors

**DOI:** 10.3390/cancers15041307

**Published:** 2023-02-18

**Authors:** Barnabas Irmer, Suganja Chandrabalan, Lukas Maas, Annalen Bleckmann, Kerstin Menck

**Affiliations:** 1Department of Medicine A, Hematology, Oncology, and Pneumology, University of Münster, 48149 Munster, Germany; 2Department of Medicine A, Hematology, Oncology, and Pneumology, University Hospital Münster, 48149 Munster, Germany; 3West German Cancer Center, University Hospital Münster, 48149 Munster, Germany

**Keywords:** liquid biopsy, extracellular vesicles, cancer, biomarker

## Abstract

**Simple Summary:**

During the past decade, liquid biopsy-related publications have increased exponentially, demonstrating the great potential of screening body fluids for diagnosis, monitoring, and prediction of therapy response in cancer patients. Next to well-established, cancer-associated analytes in the bloodstream, such as circulating tumor cells and cell-free DNA, an increasing number of studies have highlighted extracellular vesicles (EVs) as a promising source of novel tumor biomarkers. In this review, we discuss the advantages, limitations, and challenges of using EVs as cancer biomarkers, with a special focus on solid tumors.

**Abstract:**

Extracellular vesicles (EVs) are secreted by all living cells and are ubiquitous in every human body fluid. They are quite heterogeneous with regard to biogenesis, size, and composition, yet always reflect their parental cells with their cell-of-origin specific cargo loading. Since numerous studies have demonstrated that EV-associated proteins, nucleic acids, lipids, and metabolites can represent malignant phenotypes in cancer patients, EVs are increasingly being discussed as valuable carriers of cancer biomarkers in liquid biopsy samples. However, the lack of standardized and clinically feasible protocols for EV purification and characterization still limits the applicability of EV-based cancer biomarker analysis. This review first provides an overview of current EV isolation and characterization techniques that can be used to exploit patient-derived body fluids for biomarker quantification assays. Secondly, it outlines promising tumor-specific EV biomarkers relevant for cancer diagnosis, disease monitoring, and the prediction of cancer progression and therapy resistance. Finally, we summarize the advantages and current limitations of using EVs in liquid biopsy with a prospective view on strategies for the ongoing clinical implementation of EV-based biomarker screenings.

## 1. Introduction

Cells in the human body can exchange information through a variety of mechanisms. One that stands out is cell communication via extracellular vesicles (EVs) because they transport proteins, metabolites, lipids, and nucleic acids. All living cells secrete EVs, that promote not only physiological but also pathological processes, such as cancer. As a consequence, EV-based liquid biopsy can be used to gain information about tumor progression and the tumor itself. Next to other blood-derived analytes, such as circulating tumor cells (CTCs), circulating tumor DNA (ctDNA), cell-free RNA, microRNA, and long non-coding RNA (lncRNA), EVs have received more attention in recent years as a promising source of cancer biomarkers in liquid biopsies.

EVs can be categorized regarding either their cellular origin or size. EVs derived from the intracellular endosomal system are classically denoted as “exosomes”, whereas vesicles that are directly shed from the plasma membrane are known as “ectosomes” or “microvesicles”. Since this distinction was not always clear, EVs with a size up to 200 nm are nowadays simply called “small EVs” (sEVs), and vesicles larger than 200 nm are “large EVs” (lEVs) [1]. Recently, another type of nanoparticle has been described, with a size below 50 nm, the so-called exomeres. These particles are non-membranous, and their function is still unclear [2].

The shedding of lEVs from the plasma membrane is triggered by an increase in intracellular calcium (Ca^2+^) [3]. High Ca^2+^ levels lead to transient and local activation of the cysteine protease calpain, which cleaves a panel of cytoskeletal proteins and thus destabilizes the connections of the plasma membrane to the underlying cytoskeleton [4]. Combined with hydrostatic pressure, this results in the outward budding of the membrane and the formation of small blebs [5,6,7]. Proteins of the endosomal sorting complex required for transport (ESCRT) then cut off these blebs and release them as lEVs into the extracellular environment [8] (Figure 1). Furthermore, rising intracellular Ca^2+^ levels lead to an inactivation of flippases and concordant activation of floppases and scramblases. Under normal conditions, flippases keep the phospholipids phosphatidylserine (PS) and phosphatidylethanolamine (PE) in the inner leaflet of the plasma membrane, thereby controlling membrane asymmetry [5]. Their Ca^2+^-mediated inactivation consequently facilitates PS and PE externalization, and thus, their expression on the surface of blebbing lEVs.

In contrast to lEVs, sEVs are generated within late endosomes. Intraluminal vesicles (ILVs) are formed by the invagination of late endosomal membranes, which leads to the formation of multivesicular bodies (MVBs) [9]. Comparable to lEV biogenesis, this step involves ESCRT proteins, which are recruited to late endosomal membranes via the ESCRT-binding ALG-2-Interacting protein X (ALIX). In turn, ALIX recruitment was shown to be mediated by the adaptor protein syntenin, which binds to the C-terminus of syndecan heparan sulfate proteoglycans upon their endocytic uptake [10,11]. However, there is also evidence for an ESCRT-independent formation of sEVs in which tetraspanins such as CD9 or CD63 are involved [12]. As a final step, sEVs are released into the extracellular environment upon fusion of the MVBs with the plasma membrane (Figure 1).

Independent of their mechanism of formation, EVs contain components of their parental cells that can be passed on to the recipient cell by various uptake mechanisms. Described mechanisms are clathrin-dependent and -independent endocytosis, macropinocytosis, phagocytosis, and membrane fusion [13].

## 2. EVs in Cancer

As do healthy cells, tumor cells secrete EVs, and there is growing evidence that such tumor-derived EVs contribute to metastasis, angiogenesis, and chemotherapy resistance [14]. They can affect neighboring cells within the tumor microenvironment, but also distant cells and organs, where they contribute to the formation of pre-metastatic niches [15]. Some proteins, which are overexpressed in tumors, have been found to be involved in EV biogenesis. Examples are the tyrosine kinase SRC, the adapter protein syntenin, and mutated adenomatous polyposis coli (APC) [10,16,17]. These observations are consistent with increased EV secretion rates in tumor cells compared to healthy cells [18]. EVs released by tumor cells have been shown to promote angiogenesis, repress responses of the immune system, and stimulate remodeling of the extracellular matrix (ECM), thus creating a tumor-supportive microenvironment [19,20]. In addition, tumor-derived EVs can transfer drug resistance, and hence, represent promising targets for future therapy approaches [21]. These pathological effects of tumor-derived EVs are mediated by their molecular cargo, which comprises tumor-specific proteins, microRNAs, or mRNAs as well as metabolites [22,23].

Until now, CTCs and cfDNA have been the most comprehensively studied analytes in liquid biopsies for cancer biomarker detection. However, EVs also exhibit enormous diagnostic and prognostic potential in this field. In this review, we discuss the different methods for standardized EV isolation/characterization from peripheral blood, the most frequently used biofluid for liquid biopsy, and focus on the different approaches for using EVs as biomarkers for cancer.

## 3. EV Isolation and Purification from Biological Fluids

The isolation of EVs from body fluids is challenging as they are prone to contamination with non-EV proteins, lipoproteins, and high-density lipoproteins (HDL) [24]. Such contaminants interfere with the isolation of pure EVs for therapeutic, diagnostic, or prognostic use. A number of isolation methods have been described, which are summarized in Figure 2 and are discussed later in this chapter with regard to their potential applicability in clinical settings. The various EV isolation methods can be subdivided into different categories based on the physical/chemical properties exploited for isolation: centrifugation-based methods, size-based isolation, affinity-based isolation, precipitation, and recently developed microfluidic techniques. Each method has its advantages and disadvantages, and the choice of method largely depends on the intended application of the EVs [25,26]. For instance, for the quantification of established EV-based cancer biomarkers, a high-throughput isolation method with high yields but lower purity might be sufficient. On the other hand, a method providing EVs with high purity but lower yields might be preferable for identifying novel biomarkers found in EVs [27,28]. A promising approach to improve EV purity is to combine different isolation methods to achieve a contamination rate lower than that possible with one-step isolation procedures [29].

### 3.1. Centrifugation-Based Methods

In principle, centrifugation-based methods use a centrifugal force to isolate EVs based on size and density. To isolate EVs from the blood via differential ultracentrifugation, the sample is spun in a sequence of increasing speeds. The protocol usually starts with a spin at 1200× *g* to pellet the majority of blood cells. The supernatant is then spun at 2000× *g* for pelleting cell debris, residual platelets, and very large EVs. Finally, two centrifugation steps are performed at 10,000–14,000× *g* for lEVs and at 100,000× *g* to pellet sEVs [23,30,31,32]. The quantity of the collected EVs is affected by the centrifugation speed and time [33]. Furthermore, the protocol must be modified according to the source of the biofluid [34]. Although differential ultracentrifugation is still considered to be the gold standard for EV isolation, it is time-consuming, can damage the EVs, and is prone to the co-isolation of protein contaminants [35,36,37]. The isolation of lEVs can be especially challenging due to the large number of platelets or urinary sediments in some biofluids. To reduce vesicle damage and the level of impurities, EVs can be loaded onto a density gradient (i.e., iodixanol or sucrose) prior to ultracentrifugation [35,36,37,38,39]. This results in a separation of EVs in specific fractions according to their buoyant density, while protein contaminants accumulate in different fractions. To further improve EV purity, it is recommended to additionally remove the sample from HDL particles, as they are in a similar density range as EVs. In summary, centrifugation-based methods give high EV yields but low purity and require a time- and labor-consuming isolation procedure, which is not ideal for routine clinical applications.

### 3.2. Size-Based Isolation

Size-exclusion chromatography (SEC) is a well-established technique used for macromolecule separation based on size. The mobile phase containing the biofluid is loaded onto a column packed with linked porous agarose beads representing the stationary phase [28]. While EVs do not fit into the pores and pass rapidly through the column, protein aggregates do enter the pores, and thus, take longer to elute [40]. Various studies have shown the superiority of SEC over ultracentrifugation [28,41]. As the properties of the EVs are only minimally affected, SEC maintains their biological functions [28]. Furthermore, as with density ultracentrifugation, SEC reduces contaminations with HDL better than conventional ultracentrifugation since HDL particles are smaller than EVs [28]. A major drawback of this method is that it cannot efficiently separate lEVs from sEVs. However, it enables the separation of EVs from small non-EV contaminants.

Filtration is another popular size-based technique for EV isolation, in particular for large-scale EV isolation from diluted samples, such as urine [28,42]. Ultrafiltration utilizes membranes with a defined molecular weight cutoff, usually between 10 and 100 kDa [28]. EVs are retained in the filter, while smaller protein aggregates or lipoparticles pass through. Filtration is faster and easier to handle than ultracentrifugation, and only small amounts of biofluid are required [43]. However, EVs isolated by ultrafiltration suffer from considerable amounts of protein contaminants, loss of EV membrane integrity, and morphological changes, as the EVs are deformed at the filter interface due to pulling forces [44,45]. To solve this problem, tangential flow filtration (TFF), a novel size-based ultrafiltration method, was established in 2018 [46]. In TFF, a stream of biofluid flows tangentially by moving across the membrane. Particles with a molecular weight below a membrane-specific cut-off point are removed, while larger molecules, such as EVs, remain in the system and are concentrated. This method is gentler than traditional ultrafiltration and can avoid filter clogging, thus leading to higher EV yields [46].

Another frequently used size-based EV isolation strategy is field-flow fractionation (FFF), which allows for EV isolation based on the diffusion coefficient [47]. Separation occurs in a thin flow channel, in which the laminar channel flow carries the sample through the channel. Perpendicular to this channel flow, a cross-flow is applied, creating a force field, which drives the particles in the direction of the channel bottom. Due to the diffusion coefficient, small particles diffuse back into the channel faster than large particles and are thereby eluted faster than large particles [47]. In combination with light scattering detectors, FFF can provide accurate information on the EV size, morphology, and aggregation state [28]. This gentle isolation procedure is a major advantage of FFF. However, only small quantities of biofluid can be processed, thereby limiting the application when larger volumes need to be processed [28].

### 3.3. Affinity-Based Techniques

These methods are based on the highly selective and specific interactions between proteins found on the EV membrane and the corresponding receptors, e.g., antibodies. The receptors are commonly immobilized on a solid medium, such as magnetic beads or chromatography columns [28]. Affinity-based EV isolation is easy to perform, allows for single-step purification even from diluted samples, and can enrich specific EV subsets from complex biofluids. For instance, melanoma-derived sEVs were already successfully captured from plasma using magnetic beads coated with a chondroitin sulfate proteoglycan 4 (CSPG4) antibody [48]. Furthermore, well-established methods, such as enzyme-linked immunosorbent assay (ELISA), can easily be used to quantify the isolated EVs from small amounts (e.g., 100 µL) of plasma, serum, or urine [49]. A major caveat of affinity-based isolation is that it cannot exploit intravesicular antigens. Furthermore, eluted EVs can lose some of their activity and can have different characteristics than those isolated based on size. In addition, the clinical applicability of this method is doubtful as the method is expensive and the EV yield is low [49].

### 3.4. Polymer Precipitation

Another way to isolate EVs is by polymer precipitation, in which a hydrophilic polymer/reagent is added to the biofluid [50]. The polymer is able to interact with the water surrounding the EVs, thereby causing less soluble components to precipitate from the solution. The sample can then be spun down in order to obtain a collection of EVs [40]. Although this method also creates a pellet similar to ultracentrifugation, no excessive centrifugal forces are necessary [51]. Furthermore, the method is rapid and results in high EV yields. To further minimize the expenses, lower-cost precipitation reagents, such as polyethylene glycol (PEG), can be used. Since this method provides easy three-step EV isolation comprising mixing, incubation, and centrifugation, polymer precipitation is becoming more and more popular. Nevertheless, EV purity can be low as the polymer precipitates not only EVs but also any water-soluble material, including lipoproteins and nucleic acids. In addition to the problem of co-isolation, the reagents added for precipitation are difficult to remove and therefore interfere with EV properties and potential downstream applications [51,52,53].

### 3.5. Microfluidic Technology

With this novel technology, EVs are captured within micro-sized channels either by specific surface markers (microfluidics-based immunoaffinity capture) or size [54]. Although microfluidic technologies, such as Exo Chip, enable the isolation of various types of EVs and minimize contamination by proteins, they require complicated photolithography fabrication and saturation. Generally, this method integrates both EV isolation and disease detection in one platform and allows for rapid EV isolation from small sample volumes on a single chip. Major advantages are that EVs maintain their morphology, and the cost, processing time, and consumption of reagents are reduced [49]. However, as this is a recently developed method for EV isolation, little is known about its suitability for clinical applications. Thus, further establishment and improvement of microfluidic-based EV isolation are required for a subsequent translation into the clinical routine.

## 4. EV Analysis Methods

Isolated EVs can be quantified and analyzed by different methods. These methods can be used to measure basic EV characteristics (number, size, and morphology) as well as for quantitative and qualitative analysis of EV cargo (proteomic, transcriptomic, and metabolic).

EVs are commonly characterized morphologically by transmission electron microscopy. The EV number and size can be quantified using light scattering techniques, such as nanoparticle tracking analysis (NTA), or by tunable resistive pulse sensing (TRPS) [55]. NTA is a technique combining laser light scattering microscopy with a charge-coupled device (CCD) camera. Based on the Brownian motion of the particles, the software can track the distance covered by the EVs in a given time and calculate their size with the Stokes–Einstein equation [56]. TRPS forces particles through a pore via pressure and voltage. Each particle causes a resistive pulse signal that is captured and measured [57]. Nevertheless, data from NTA and TRPS need to be considered cautiously, as parameters such as pore sizes and detection sensitivity influence the measurement when analyzing EV populations [58]. The detection with NTA-based analysis is influenced by two parameters: camera-level and detection threshold [59]. Higher camera levels lead to a brighter appearance of the particles, which results in increased detection of weak-scattering proteins. Additionally, an increased detection threshold consequently results in a higher number of detected particles [59]. For TRPS-based quantification, the sensitivity can be variable based on the nanopores used. The sensitivity with different nanopore setups is based on the size of the smallest detectable particle. As such, for EV samples with an unknown size distribution, it is more difficult to obtain the optimal size range to detect all particles [59]. A major advantage of TRPS measurement over the NTA methodology is the possibility to spike biological fluids with polystyrene beads of a known size and concentration to improve the accuracy of EV quantification with TRPS [60]. This method of calibration is not suitable for NTA analysis, as the methodology does not discriminate between EVs and the calibration beads [61].

The evaluation of EV-associated biomarker expression requires methods that specifically detect intravesicular or membrane-associated EV cargo. The expressions of protein-based biomarkers within or on the surface of EVs can be assessed using flow cytometry, ELISA, immunoblotting, or mass spectrometry. The latter two methods are more suitable for the identification of novel EV-based biomarkers than high-throughput screening approaches. While lEVs were already shown to be suitable for the validation of several known cancer biomarkers via flow cytometry [23], similar readouts concerning sEVs remain challenging due to their small size. Compared to conventional flow cytometry, which has a resolution limitation for particles < 300 nm, high-resolution and imaging flow cytometers have been demonstrated to be promising for sEV protein biomarker quantification and subtype characterization [62]. In addition, sEVs can be coupled to larger latex or magnetic beads, which enables bead-assisted flow cytometry [63]. According to the Minimum Information about a Flow Cytometry Experiment (MIFlowCyt) guidelines, relevant steps need to be observed for flow cytometry-based EV analysis, ranging from sample preparation and assay controls to technical guidelines [64]. Another method to quantify the amount of one or more specific proteins in EV samples is ELISA [65]. This method allows for the rapid analysis of specific proteins either generally found on EVs (such as tetraspanins) or tumor-specific antigens, although in a size-independent manner [66].

In addition to the proteomic cargo, EVs can also be analyzed for their metabolic profile using mass spectrometry. In the study performed by Buentzel et al., lEVs derived from the plasma of breast cancer patients or healthy controls were distinguishable based on their distinct metabolome [22]. Furthermore, the metabolomic profiling of lEVs also allowed for patient allocation to the distinct molecular breast cancer subtypes.

A number of approaches can be used to analyze the nucleic acid cargo found in EVs (e.g., DNA, RNA, and microRNA). Dye-assisted analysis of RNA content in EVs was described previously [67]. However, some dyes might also detect non-EV-associated RNA. Thus, real-time polymerase chain reaction (PCR) and next-generation sequencing (NGS) might be more promising analytical techniques for nucleic acid detection and quantification in EVs [68]. Particularly, EV-associated microRNAs are an emerging field of interest, as some microRNAs are tissue- and cell-specific and, as such, might represent the pathophysiological state of the cell of origin more precisely than the proteomic cargo [69,70,71].

To conclude, many different techniques for the quantitative and qualitative analysis of EV number, size, and cargo have been successfully applied for liquid biopsy analysis. However, their clinical implementation still requires standardized protocols that comply with the Minimal Information for Studies of Extracellular Vesicles (MISEV). These guidelines were designed to minimize the influence of different detection and analysis methods as well as confounding factors, including co-isolated non-vesicular contaminants, to pave the way for EV-associated cancer biomarkers in liquid biopsies.

## 5. EV-Associated Cancer Biomarkers—Translational Studies and Clinical Applications

Until now, multiple translational studies have addressed the question of whether EV levels in the blood of cancer patients or their molecular cargo are suitable for reflecting clinical parameters such as tumor burden, disease progression, or therapy response. Based on these publications, the following chapter focuses on circulating EV-based biomarkers for a variety of solid tumor entities, which have been identified and characterized in a clinical setting. We have not referenced the literature on the importance of EV-based liquid biopsies in hematological malignancies, but this was comprehensively reviewed by others [72].

### 5.1. Circulating EV Levels in Plasma

The currently available data indicate an increase in sEV abundance in blood samples from patients suffering from various cancer entities compared to healthy donors, as shown, e.g., for melanoma [73], glioblastoma multiforme (GBM) [74], prostate cancer (PC) [66], head and neck cancer (HNC) [75], breast cancer (BC) [76], and colorectal carcinoma (CRC) [77]. The latter two studies showed higher sEV plasma levels in cancer patients to be associated with therapy resistance, disease progression, and shortened overall survival. On the other hand, a study from 2017 found that patients with various types of solid tumors exhibited similar levels of lEVs in their plasma as healthy controls [23]. Different EV isolation protocols and a lack of standardization for EV classification procedures might be the cause for such variations.

The cellular origin of the increased number of EVs in blood from cancer patients is still under debate. Some evidence points to the primary tumor mass as the main driver for exaggerated EV secretion, seeing that markedly lower EV levels have been detected in post-operative plasma samples [74,75,78]. Hence, the measurement of plasma EV concentrations holds promise for monitoring tumor burden in cancer patients after surgery. Chemo- and radiotherapy have triggered EV secretion in patients suffering from different cancer entities [79,80,81]. This observed effect might help to improve the evaluation of therapy responses in the future. Measurements of total plasma EV protein might be a relevant alternative to the quantification of EV concentrations, as the former might reflect both aberrant EV numbers and deregulated EV cargo loading in cancer patients. Indeed, increased EV protein levels were detected in the blood of HNC patients with recurrence [82]. However, numerous other factors are known to influence EV numbers in the blood. Elevated plasma EV concentrations have been found in physiological conditions, such as physical exercise [83] and pregnancy [84], and in pathological processes, including ischemic stroke [85,86,87], Crohn’s disease [86], and diabetes [87]. The diagnostic potential of measuring all EVs present in the blood without further attribution to their cellular origin is therefore likely to be prone to misinterpretations. One way of circumventing these limitations is represented by analyzing the molecular cargo of plasma-derived EVs.

### 5.2. Molecular Cargo of Circulating EVs in Plasma

#### 5.2.1. Tumor-Specific EV Biomarkers for Cancer Diagnostics

In the field of EV-based liquid biopsy proteins, DNA and RNA are the most comprehensively studied biomarkers for cancer detection, monitoring, and prediction. In addition, EV-associated metabolites are increasingly being discussed as promising biomarkers [22]. Tumor-derived EVs have been shown to reflect the molecular composition of the secreting cancer cells in several studies; by proteomic evaluation of tumor-derived EVs with concomitant hierarchical clustering, EV samples were segregated according to their cellular origin [88]. Similar observations have been made for the RNA and DNA content of EVs secreted by different cancer entities [89,90,91,92]. However, the expression rates of EV-associated cancer biomarkers within plasma-derived liquid biopsy samples do not necessarily reflect the molecular situation of the corresponding tumor because of the large amount of non-tumor EVs in the blood. Due to this, reports about increasing the plasma levels of EV-associated biomarkers in cancer patients alone cannot exclude non-tumor tissue as a driver for these observations [93,94].

A direct strategy for the identification of cancer-specific EV biomarkers is the molecular comparison of plasma EVs with matched tumor tissue biopsies. Based on this approach, Sun et al. demonstrated that vesicular copine-3 (CPNE3) expression in the plasma samples of CRC patients positively correlated with the protein signal from the corresponding tumor tissue and with overall survival [95]. However, the vesicular expression of cancer biomarkers does not always correlate with tumor tissue levels, as was shown for the NH2-terminally truncated P73 (ΔNP73) mRNA [96]. Here, the observed increase in ΔNP73 mRNA expression within plasma-derived sEVs from CRC patients might be a systemic rather than a tumor-driven response. For CD82 even a negative correlation between primary breast cancer and circulating sEV expression have been observed [97]. A study in 2020 took a step further and analyzed the proteomes of sEVs in plasma and those of primary tumor and tumor-adjacent tissue samples from pancreatic and lung cancer patients [98]. When compared to healthy plasma samples, 51 and 19 sEV proteins were exclusively identified in pancreatic and lung cancer patients, respectively. Sixteen of the EV proteins linked to pancreatic cancer were only detected in biopsy samples from the primary tumor and not in tumor-adjacent tissue, which strengthens their usability as cancer-specific biomarkers. Indeed, machine-learning classification of plasma EV cargo detected cancer with a sensitivity and specificity above 90% and was able to differentiate among different tumor entities, which makes analyzing tumor-EV cargo a powerful tool in cancer diagnostics.

Several studies have reported detection sensitivities of 100% after analyzing circulating sEVs regarding their expression of biomarkers, such as miR-1246 for breast cancer [99] and prostate-specific antigen (PSA) for prostate cancer [100]. PSA additionally performed well as an early detection EV biomarker in stage I cancer patients, as did survivin for prostate cancer [101] and Del-1 for breast cancer [102]. In pancreatic ductal adenocarcinoma (PDAC) patients, an EV-associated expression of highly upregulated in liver cancer (HULC) long non-coding RNA (lncRNA) was demonstrated to perform even better than the already established circulating serum biomarkers carbohydrate antigen 19-9 (CA19-9) and carcinoembryonic antigen (CEA) in terms of discriminating between cancer patients and non-PDAC controls [103]. Not only cancer-specific but also pan-cancer biomarkers have been demonstrated to be diagnostically relevant, as has been shown for miR-21, which has been detected at high levels in circulating sEVs of pancreatic, breast, and lung cancer patients [104,105,106]. In these studies, high vesicular miR-21 expression was also associated with shortened survival and chemoresistance. Similarly, extracellular matrix metalloproteinase inducer (EMMPRIN) was detected in high levels in plasma lEVs from cancer patients with different kinds of solid tumors and showed a robust detection efficiency, especially in combination with epithelial cell adhesion molecule (EpCAM), mucin-1 (MUC1), and epidermal growth factor receptor (EGFR) expression rates [23]. The detection of cancer-specific mutations using EV-associated DNA represents another powerful tool for cancer diagnostics and is described in the following section.

#### 5.2.2. EV-Associated DNA for Mutation Screenings

Recently, an increasing number of studies have addressed the applicability of EV-associated DNA in blood for the detection of cancer-specific mutations, especially in comparison with cfDNA. Data from 2017 indicate that the detection of Kirsten rat sarcoma (KRAS) mutations in PDAC patients via plasma-derived EV-DNA was more sensitive than cfDNA-related detection [107] and additionally predicted patient survival. This finding was confirmed in a later study for KRAS mutations in CRC patients [108]. In turn, Thakur et al. reported similar detection rates when using either circulating EV-DNA or cfDNA for the identification of several mutations in CRC patients [109]. In the future, however, a combination of both EV-derived nucleic acids and cfDNA might represent the most efficient procedure for non-invasive cancer-associated mutation screenings [110,111]. This approach was also applied for the development of a clinical test developed by Exosomes Diagnostics (ExoDx Lung(EGFR T790M)), which efficiently identified different EGFR mutations (L858R, T790M, and exon 19 indels) in plasma sEVs from a large cohort of non-small cell lung carcinoma (NSCLC) patients [112]. Measuring the mutation status of circulating EVs can potentially allow one to monitor parameters such as tumor size [109] and stage [113]. Notably, in the latter study, GBM-associated isocitrate dehydrogenase 1 (IDH1) mutations were detected even in low-grade glioma patients with an intact blood–brain barrier, confirming the high sensitivity of EV-based mutation screenings for cancer diagnostics.

Due to their high diagnostic impact, circulating EV-based biomarkers are also considered to efficiently represent the remaining tumor cells in the blood of cancer patients after therapy. For the monitoring of this so-called minimal or molecular residual disease (MRD) via liquid biopsies, the recent literature has often focused on ctDNA and CTCs [114,115]. However, more and more studies, which are outlined in the following section, have addressed and confirmed the usability of plasma-derived EVs for disease monitoring and relapse prediction.

#### 5.2.3. Therapy Monitoring

In order to investigate the potential of tumor-specific EV biomarkers in the circulation for therapy monitoring, researchers have compared expression levels in matched patient samples before and after therapy. Several protein- and RNA-based biomarkers that were elevated in the plasma-EVs of cancer patients did show reduced expression levels after primary tumor resection or chemotherapy. Some of these proteins with reduced expression after surgery are EMMPRIN [116], 60 kDa heat shock protein (HSP60) [117], and glypican-1 (GPC1) [118] for CRC, cytoskeleton-associated protein 4 (CKAP4) for PDAC [119], and prostate-specific membrane antigen (PSMA) for prostate cancer [120]. Decreased levels of EVs positive for EGFR and its oncogenic variant EGFRvIII were detected in GBM patients after chemotherapy [121]. Similarly, EV-associated RNA molecules, such as the lncRNA HOX antisense intergenic RNA (HOTAIR) and the microRNAs (miR) miR-301 and miR-155, have been proven to be suitable markers for monitoring the response to surgery and chemotherapy, respectively, in breast cancer patients [122,123]. For some markers, even rising expression rates on circulating EVs can reflect a successful response to certain therapies. For example, increasing levels of EpCAM-positive EVs in the plasma of PDAC patients during chemotherapy were associated with longer progression-free survival [124]. This short-term increase in EV-associated EpCAM possibly reflects a therapy-induced stress response of the tumor.

Within the past decade, T-cell-targeted immunomodulators directed against the immune checkpoints of cytotoxic T-lymphocyte-associated protein 4 (CTLA-4), programmed cell death 1 (PD-1), and its ligand (PD-L1) have revolutionized cancer treatment for several types of tumors [125]. Since the precise identification of responding and non-responding cancer patients before or during certain immunotherapies remains challenging, circulating EV cargo was tested for its ability to predict and monitor therapy responses after immune checkpoint inhibitor treatments. Elevated PD-L1 expression on plasma sEVs during anti-PD-1 immunotherapy was linked to successful therapy response in melanoma patients [126], and high PD-L1 mRNA expression on circulating sEVs from melanoma and NSCLC patients before anti-PD-1 treatment was correlated with partial or complete remission after therapy [127]. Vesicular miR-320b, -c and –d, in turn, seem to be increasingly expressed in the blood of NSCLC patients identified as non-responders to antiPD-1 therapy [128]. 

#### 5.2.4. Prediction of Disease Progression and Therapy Resistance

Apart from the promising application of EV-based liquid biopsy for cancer detection and the monitoring of therapy responses, the prediction of disease progression and resistance to certain therapies represents another area for its clinical use. The majority of EV biomarkers associated with chemoresistance exhibited high plasma levels in the blood of non-responding cancer patients as shown for several RNA molecules [129,130] and protein-based markers [131,132]. However, other studies also identified significantly downregulated levels of EV-associated miRNAs in the circulation of non-responders to chemo- and radiotherapy [133,134].

Regarding the potential use of plasma-derived EVs in the prediction of disease progression, miR-1247-3p for hepatocellular carcinoma (HCC) and the entire mRNA profile for osteosarcoma were shown to be associated with metastasis formation [135,136], while miR-17-5p and miR-92a-3p for CRC and miR-638 for HCC were correlated with cancer stage [137,138]. The expressions of miR-217 and lncRNA colorectal neoplasia differentially expressed (CRNDE) were linked to both metastasis and stage [139]. In addition, high plasma levels of Treg-derived CD3 and PD-L1 double-positive sEVs predicted recurrence in HNC patients after a combination therapy [82], which underlines the importance of also considering non-tumor-derived EV biomarkers for the prediction of disease states and therapy outcome. Figure 3 gives an overview of most of the biomarkers described above. All given information is additionally listed in the Appendix A.

#### 5.2.5. Commercially Available Tests Exploiting EVs as Cancer Biomarkers

In spite of the large number of potential biomarkers for EV-driven cancer diagnostics, only three clinical tests that exploit EVs from liquid biopsies for cancer patient stratification have been commercially launched. All were developed by the Clinical Laboratory Improvement Amendment (CLIA)-certified laboratories of Exosome Diagnostics. ExoDx™ Lung (ALK) and ExoDx Prostate IntelliScore (EPI) were designed for the quantification of certain sEV-associated mRNAs as biomarkers for NSCLC and prostate cancer, respectively. The ExoDx™ Lung (ALK) test detects EML4-ALK mutations via quantitative PCR analysis of plasma EVs from NSCLC patients with 88% sensitivity and 100% specificity [141]. In turn, the EPI test uses urine as an EV source and is described in the following section. Furthermore, Exosome Diagnostics recently announced MedOncAlyzer 170 as a test, which also utilizes a combination of blood-derived EV-RNA/DNA and cfDNA for detecting a panel of mutations in multiple cancer entities [111].

### 5.3. Clinical Relevance of EVs from Other Body Fluids

In addition to blood as an EV-based liquid biopsy source, other body fluids hold great promise for the diagnosis, prognosis, and monitoring of cancer patients as well. One of the first commercially available tests using sEVs for cancer patient stratification is urine-based and called ExoDx Prostate IntelliScore (EPI), which detects vesicular mRNA levels of prostate cancer antigen 3 (PCA3), ETS transcription factor ERG (ERG), and SAM-pointed domain containing ETS transcription factor (SPDEF). In clinical studies, the EPI test has been demonstrated to efficiently discriminate high-grade (Gleason score ≥ 7) from low-grade (Gleason score 6) prostate cancer and benign disease in men ≥ 50 years with equivocal PSA levels (2–10 ng/mL), thereby reducing the number of unnecessary tissue biopsies [142,143]. In general, urine-derived EVs have been investigated in multiple studies for the assessment of urological malignancies such as prostate, bladder, and kidney cancers [144].

Further examples of the predictive ability of EV-based cancer biomarkers in other body fluids are vesicular miR-21 in the cerebrospinal fluid of GBM patients, which was associated with poor prognosis and tumor recurrence [145], and EGFR mutations detected by the EV-DNA of NSCLC-derived bronchial wash fluid, which was correlated with disease progression [146]. A study in 2016 observed that even epigenetic modifications of EV-DNA can function as clinically relevant biomarkers for cancer detection. The researchers used gastric juice samples for the quantification of EV-associated BarH, such as homeobox 2 (BARHL2) DNA methylation, which discriminated gastric cancer (GC) patients from non-GC controls with 90% sensitivity and 100% specificity [147]. Additionally, salivary gland fluid seems to be a promising source for EV-based liquid biopsies with regard to oral cancer, although it is prone to contamination with bacterial EV [148].

For the detection of cholangiocarcinoma, claudin-3 was reported to be enriched on bile-derived EVs, showing a sensitivity and specificity of 87.5% [149]. Consistently, this tight junction protein was previously shown to be differentially expressed in tumor cells from CRC patients, leading to the hyperactivation of Wnt-signaling [150].

## 6. Advantages and Limitations of EVs as Liquid Biopsy Biomarkers

Despite their underrepresentation in liquid biopsy-based cancer diagnostics, EVs are undoubtedly suitable for non-invasive biomarker detection processes. Similar to CTCs, they can be used as a multi-analyte platform for the detection and quantification of different kinds of cancer-related molecules. Whereas CTCs suffer from rather low numbers in blood, EVs are highly abundant in all human body fluids and provide a wide range of possibilities for their diagnostic use [140]. EV cargo is also less prone to degradation due to the protective function of the vesicular lipid membranes, and is, hence, more stable than tumor-associated cfDNA and soluble proteins in body fluids [151]. Probably as a result of the combination of their abundance and their cargo protection, EVs have shown high detection sensitivities and have frequently been observed to be even superior to detection strategies involving cfDNA or soluble proteins as tumor antigens [100,103,107]. EV-based liquid biopsies have shown convincing results in clinical studies, particularly for early cancer diagnostics and conclusive monitoring of minimal-residual disease in follow-up patients after therapy [100,101,102]. However, their translation into clinical practice still faces several obstacles. Since most current characterization techniques use bulk analyses of the EVs, which comprise multiple subpopulations from many different cellular sources, the resulting parameters, such as the concentration or expression profiles, are prone to biological variations. This is further aggravated by the lack of standardized EV isolation and characterization protocols. The effects of pre-analytical variables, such as the storage time, the number of freeze-and-thaw cycles, conditions during transport, etc., must be seen as sources of technical variations. Furthermore, most clinical studies have been performed with rather small patient cohorts, which weakens the power of the studies and the reliability of the respective observations.

Modern approaches that provide EV quantification and molecular characterization on a single particle level, such as advanced imaging flow cytometry or fluorescence-based NTA, could help to improve EV classification, thereby reducing biological variations [152,153]. As a solution for the technical variations, the International Society for Extracellular Vesicles (ISEV) released the MISEV guidelines in order to lay the foundation for achieving comparability of EV-based studies using different protocols [1]. Moreover, EV characterization protocols for the direct analysis of body fluids without prior isolation steps represent a promising strategy to achieve standardized high-throughput screenings for cancer diagnostics. As an example, whole-blood EV characterization using alternating current electrokinetic (ACE) microarray chips with subsequent on-chip immunofluorescence analysis was shown to be suitable for detecting PDAC via GPC1 and CD63 expression [154]. Taken together, EV-based liquid biopsy holds great promise for cancer diagnosis, prognosis, and therapy monitoring but still requires precise EV subtype classification, standardized protocols, and larger patient cohorts for its further clinical translation.

## 7. Conclusions

EVs have emerged as critical mediators of intercellular communication in cancer. Tumor-EVs circulating in body fluids, such as blood, harbor the malignant traits of their cells of origin, which makes them ideal biomarker candidates for detecting, assessing, and monitoring tumor growth in liquid biopsies. In this context, not only tumor-EVs but also EVs derived from non-malignant cells reacting to the growing tumor contribute to the vesiculome measurable in body fluids and can provide valuable information. However, the use of EVs as cancer biomarkers in routine diagnostics is still hampered by the lack of standardization, the lack of data from large patient cohorts, and methodological challenges. Once these problems have been solved, EVs have great potential to become valuable tools in liquid biopsies, either alone or in combination with other body fluid components.

## Figures and Tables

**Figure 1 cancers-15-01307-f001:**
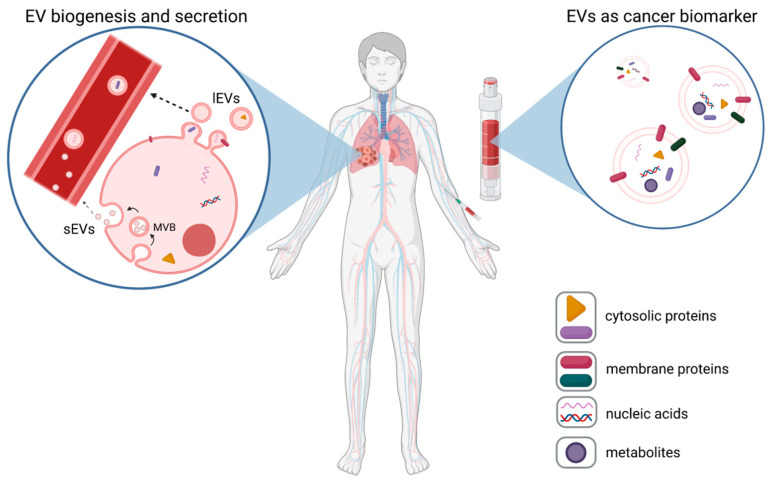
EVs as cancer biomarkers. Tumor cells release high numbers of plasma membrane-derived large EVs (lEVs) and endosomal-derived small EVs (sEVs)), which are released from multi-vesicular bodies (MVBs) into body fluids, such as blood (**left panel**). These tumor-EVs can be assessed via liquid biopsies (**right panel**), and based on the expression of tumor-specific EV proteins, nucleic acids, or metabolites, they can be used as biomarkers for cancer (exemplarily shown for lung cancer).

**Figure 2 cancers-15-01307-f002:**
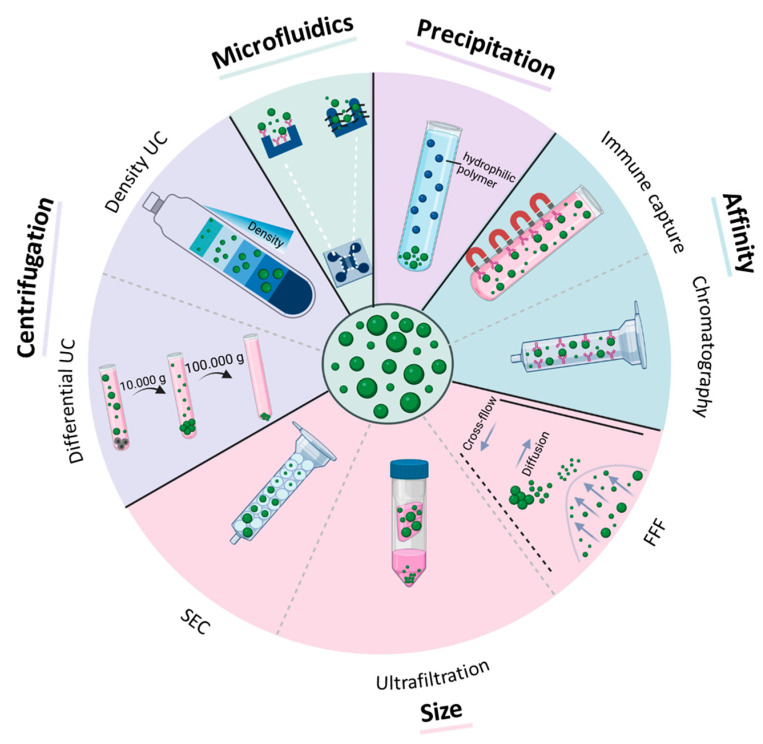
Methods for EV isolation from biofluids. Purification methods are based on centrifugal force (differential and density ultracentrifugation (UC)), size (size-exclusion chromatography (SEC), ultrafiltration, field-flow fractionation (FFF)), affinity to certain antigens (immune capture, affinity chromatography), precipitation, and microfluidic techniques (size-based microfluidic, immune capture microfluidic).

**Figure 3 cancers-15-01307-f003:**
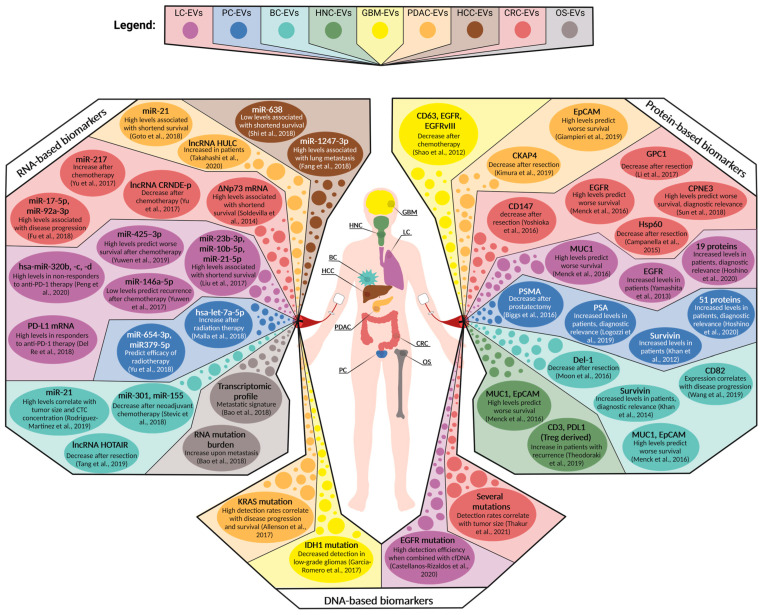
Overview of circulating EV-associated biomarkers for selected solid tumor entities tested in a clinical setting. Depicted are the protein-, RNA-, and DNA-based biomarkers that have been clinically assessed in patient cohorts suffering from glioblastoma multiforme (GBM), head and neck cancer (HNC), lung cancer (LC), breast cancer (BC), hepatocellular carcinoma (HCC), pancreatic ductal adenocarcinoma (PDAC), colorectal carcinoma (CRC), prostate cancer (PC), and osteosarcoma (OS) [23,80,82,93,94,95,96,97,98,99,100,101,102,103,104,105,106,107,109,112,116,117,118,119,120,121,122,123,124,127,128,130,133,134,135,136,137,138,139,140].

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
