# Peer review of "Extracellular Vesicles in Liquid Biopsies as Biomarkers for Solid Tumors"

_cancers, 2023, doi:10.3390/cancers15041307_

Round 1

Reviewer 1 Report

Thank you for your work, yet there are some flaws to be addressed.

- the manuscript should be revised by a native English speaker.

- The paper is well written, yet I would appreciate a critical point of view on the utilization of EVs in order to monitor minimal residual disease. Please read these papers and comment them PMID: 36596401  PMID: 36497493

Author Response

Dear reviewer,

Thank you for the helpful revision of our manuscript. We have read your comments carefully and improved our manuscript based on your suggestions. Below, you find our point-to-point responses.

- The manuscript should be revised by a native English speaker.

We thank the reviewer for the valid point and have sent the manuscript for proofreading by an English native speaker. All corrections have been implemented in the revised version of the manuscript.

- The paper is well written, yet I would appreciate a critical point of view on the utilization of EVs in order to monitor minimal residual disease. Please read these papers and comment them PMID: 36596401  PMID: 36497493

According to the reviewer’s suggestion, we implemented the utilization of EVs for MRD monitoring and the mentioned reviews in chapter 5.2.2 to summarize the diagnostic impact of circulating EVs on the one hand and lead towards the following section focusing on therapy monitoring on the other hand. We think that a critical point of view regarding the challenges and limitations of EV-based biomarker screenings also in the context of MRD is covered well in chapter 6.

Sincerely yours,

Kerstin Menck

Reviewer 2 Report

The manuscript is well-documented and nicely presented. I have sone suggestion that must be included in the manuscript:

The author must include the role of tight junction protein in the secretion of extracellular vesicles for example  claudin-2 expression increases in cancer and it also potentiates the permeability (PMID 12432083) and loss of claudin-3 induces the loss of tight junction which potentially the metastasis (PMID 28783170). 

Author Response

Dear reviewer,

Thank you for the helpful revision of our manuscript. We have read your comment carefully and improved our manuscript based on your following suggestion:

"The author must include the role of tight junction protein in the secretion of extracellular vesicles for example  claudin-2 expression increases in cancer and it also potentiates the permeability (PMID 12432083) and loss of claudin-3 induces the loss of tight junction which potentially the metastasis (PMID 28783170)."

We followed this interesting advice and investigated the role of tight junction protein expression on EVs in the context of biomarker screenings with liquid biopsies. We included a study from 2021 (PMID: 33441949), which highlights claudin-3 as a potential biomarker for cholangiocarcinoma on bile-derived EVs in chapter 5.3. Additionally, we added a reference to one of the studies you suggested (PMID: 28783170), in order to emphasize the role of claudin-3 in cancerogenous processes.

Sincerely yours,

Kerstin Menck

Reviewer 3 Report

The article on the review of cancer biomarkers in liquid biopsies is well written. Although a little editing is still required.

Line 40. Decipher the abbreviation (ctDNA). Add other cancer biomarkers in liquid biopsies such as cell-free RNA, microRNA, lncRNA.

Figure 1 - authors should decipher the abbreviation «MVBs».

If a specialized site was used to create the drawings, then a link to it should be indicated.

Line 111. Rephrase this sentence “Generally, the different methods can be divided into different…”

Chapter 3 contains few references. Authors are encouraged to add references to primary sources.

 Figure 2 - extracellular vesicles are shown in red and look like blood drops. I suggest changing the color. The inscriptions in the figure are small, unreadable, it is necessary to increase the font. And location of the tube in the “Density UC” area looks upside down.

Lines 233-242. The authors should provide comparative analysis of NTA and TRPS methods by describing advantages and limitation of each of these methods.

Line 261. The original research as an example of tumor-specific antigens analysis is missing.

Line 299-300. I would suggest to add “different detection and analysis methods”.

Lines 410-411. The point is not clear. Rephrase/expand this and add a reference.

Figure 3. Organize this information in a table as well, as supplementary information. Since the text is small and hard to read.

Author Response

Dear reviewer,

Thank you for the helpful revision of our manuscript. We have read your comments carefully and improved our manuscript based on your suggestions. Below, you find our point-to-point responses.

Line 40. Decipher the abbreviation (ctDNA).

We thank the reviewer for the suggestion and explained the abbreviation ctDNA as circulating tumor DNA in the revised manuscript.

Add other cancer biomarkers in liquid biopsies such as cell-free RNA, microRNA, lncRNA.

As requested by the reviewer we added the additional cancer biomarkers including cell-free RNA, microRNA, long non-coding RNA in chapter 1 of the revised manuscript.

Figure 1 - authors should decipher the abbreviation «MVBs».

We thank the reviewer for the point and as suggested deciphered the abbreviation MVB in the figure legend.

If a specialized site was used to create the drawings, then a link to it should be indicated.

All figures were created by the authors with BioRender. The reference according to the citing recommendations provided by the website is mentioned in the acknowledgement section.

Line 111. Rephrase this sentence “Generally, the different methods can be divided into different…”

We agree with the reviewer and rephrased the indicated sentence to: “Generally, the various EV isolation methods can be subdivided based on the physical/chemical properties exploited for isolation: centrifugation-based methods, size-based isolation, affinity-based isolation, precipitation and the recently developed microfluidic techniques.”

Chapter 3 contains few references. Authors are encouraged to add references to primary sources.

We thank the reviewer for this excellent point and have added more references in chapter 3.

Figure 2 - extracellular vesicles are shown in red and look like blood drops. I suggest changing the color. The inscriptions in the figure are small, unreadable, it is necessary to increase the font. And location of the tube in the “Density UC” area looks upside down.

We thank the reviewer for the excellent suggestion. We have changed the color of the extracellular vesicles in the figure and increased the font of the descriptions in the figure. We agree that the tube for the density UC should be turned around as it is more intuitive.

Lines 233-242. The authors should provide comparative analysis of NTA and TRPS methods by describing advantages and limitation of each of these methods.

We agree with the reviewer that the two methods for EV quantification should be compared regarding their principles and disadvantages. As requested we have added the information about the technical limitations of both methods and extracted some advantages of each of the methods over the other.

Line 261. The original research as an example of tumor-specific antigens analysis is missing.

We thank the reviewers for the excellent point and added a research paper, focusing on the detection of EVs using an ultrasensitive ELISA assay.

Line 299-300. I would suggest to add “different detection and analysis methods”.

We thank the reviewer for the excellent point and rephrased the sentence to include the difficulties in standardization not only in isolation but also in detection and analysis of EVs.

Lines 410-411. The point is not clear. Rephrase/expand this and add a reference.

We appreciate this helpful advice and decided to delete this sentence instead of rephrasing it, since its content is not matching with the chapter`s topic “therapy monitoring” but rather with “prediction of therapy resistance”.

Figure 3. Organize this information in a table as well, as supplementary information. Since the text is small and hard to read.

We agree with the reviewer and created a supplementary table with all of the information given in figure 3.

Sincerely yours,

Kerstin Menck